# Latent transition analysis of cardiac arrest patients treated in the intensive care unit

**Lifeng Xing[1], Min Yao[2], Hemant Goyal[3], Yucai Hong[1]\*, Zhongheng Zhang [1,4]\***

**1** Department of Emergency Medicine, Sir Run Run Shaw Hospital, Zhejiang University School of Medicine, Hangzhou, China, **2** Department of Surgery, Wound Care Clinical Research Program, Boston University School of Medicine and Boston Medical Center, Boston, Massachusetts, United States of America, **3** Department of Internal Medicine, Mercer University School of Medicine, Macon, Georgia, United States of America, **4** Key Laboratory of Emergency and Trauma, Ministry of Education, College of Emergency and Trauma, Hainan Medical University, Haikou, China

\* zh_zhang1984@zju.edu.cn (ZZ); zrhyc@hotmail.com (YH)

## Abstract

### Background and objective

Post-cardiac arrest (CA) syndrome is heterogenous in their clinical presentations and outcomes. This study aimed to explore the transition and stability of subphenotypes (profiles) of CA treated in the intensive care unit (ICU).

### Patients and methods

Clinical features of CA patients on day 1 and 3 after ICU admission were modeled by latent transition analysis (LTA) to explore the transition between subphenotypes over time. The association between different transition patterns and mortality outcome was explored using multivariable logistic regression.

### Results

We identified 848 eligible patients from the database. The LPA identified three distinct subphenotypes: *Profile 1* accounted for the largest proportion (73%) and was considered as the baseline subphenotype. *Profile 2* (13%) was characterized by brain injury and *profile 3* (14%) was characterized by multiple organ dysfunctions. The same three subphenotypes were identified on day 3. The LTA showed consistent subphenotypes. A majority of patients in *profile 2* (72%) and *3* (82%) on day 1 switched to *profile 1* on day 3. In the logistic regression model, patients in *profile 1* on day 1 transitioned to profile 3 had worse survival outcome than those continue to remain in *profile 1* (OR: 20.64; 95% CI: 6.01 to 70.94; p < 0.001) and transitioned to *profile 2* (OR: 8.42; 95% CI: 2.22 to 31.97; p = 0.002) on day 3.

### Conclusion

The study identified three subphenotypes of CA, which was consistent on day 1 and 3 after ICU admission. Patients who transitioned to profile 3 on day 3 had significantly worse survival outcome than those remained in profile 1 or 2.

**Data Availability Statement:** Data cannot be shared publicly because the data are owned by a third party and authors do not have permission to share the data. Data are available from the Beth Israel Deaconess Medical Center Institutional Data

Access (via https://physionet.org/content/mimiciii/1.4/) for researchers who meet the criteria for access to confidential data.

**Funding:** Z.Z. received funding from Yilu "Gexin" - Fluid Therapy Research Fund Project (YLGX-ZZ-2020005), Health Science and Technology Plan of Zhejiang Province (2021KY745), the Key Laboratory of Tropical Cardiovascular Diseases Research of Hainan Province (Grant.KLTCDR-202001) and Key Laboratory of Emergency and Trauma (Hainan Medical University), Ministry of Education (Grant.KLET-202017). Y.H. received funding from Key Research & Development project of Zhejiang Province (2021C03071). The study was funded by clinical research foundation of Zhejiang Medical Association (2019ZYC-A87).

**Competing interests:** The authors have declared that no competing interests exist.

# Introduction

Cardiac arrest (CA) is an important public health problem accounting for approximately 500,000 deaths annually in the Europe and the USA [1, 2]. A significant number of patients will survive the acute event and require post-resuscitation care in the intensive care unit after return of the spontaneous circulation (ROSC) [3]. The improvement in survival outcome is relatively small in spite of recent advances in post-CA care [4]. Therapeutic interventions for post-resuscitation care including neuromuscular blockade and inhaled Xenon have been explored [5–8]. In particular, targeted temperature management (TTM) has been shown to improve survival and neurological functions in patients with CA [9–11]. Exploration of heterogeneity of the study population can further improve the efficacy of these clinical trials. The heterogeneity of critical care syndromes such as acute respiratory distress syndrome (ARDS) and sepsis has been well studied and some subphenotypes of these syndromes have been identified, exhibiting distinct clinical presentations, clinical outcomes and responses to therapeutic interventions [12]. For example, Calfee and colleagues identified 3 subphenotypes of ARDS which responded differently to fluid management strategy [13, 14]. Gårdlund B and colleagues identified 6 subphenotypes of septic shock that showed distinct clinical characteristics [15]. In addition, the subphenotype transition has also been widely investigated because unraveling the transition pattern can have significant clinical and research implications [16–18]. For example, subphenotype stability over time can help to design trials and/or therapeutics. Subphenotype transition is also important to the question on whether difference in clinical presentation is dependent on the timing of measurement [16].

Similarly, CA patients in the ICU also exhibit significant heterogeneity [19, 20], and subphenotypes exploration may help to identify patients who might benefit most from certain therapeutics. Our previous work has identified subphenotypes of CA using cross-sectional data on the first day of ICU entry [21]. However, it is largely unknown whether the subphenotypes are stable or subject to transitions and how could this transition inform clinical decisions. Other studies show that subphenotype transition can have significant clinical implications [17, 22]. Thus, the current study aimed to characterize the latent transition pattern of CA patients by using latent transition analysis (LTA). The differences in the mortality outcome for patients with different transition paths were also explored.

# Materials and methods

## Study setting and population

The study used a large critical care database called MIMIC-III (Medical Information Mart for Intensive Care [23]. Detailed introduction of the database could be found at the website: https://mimic.physionet.org/. Briefly, the MIMIC-III is a critical care database comprising deidentified patients' data for more than 40,000 admissions who stayed in the ICUs of the Beth Israel Deaconess Medical Center. The database includes patient information such as vital sign recordings, demographics, medications, laboratory test results, imaging reports, procedures, caregiver notes, and mortality outcome. The study was conducted by utilizing anonymized database with pre-existing institutional review board (IRB) approval. The study was conducted in accordance to the REporting of studies Conducted using Observational Routinely-collected health Data (RECORD) Statement [24].

## Subjects selection

Subjects with the diagnosis of cardiopulmonary resuscitation (ICD-9 code: 9960 and 9963), cardiac arrest (ICD-9 code: 4275) and ventricular fibrillation (ICD-9 code: 4274) were

screened for potential eligibility [25]. Only the first admission was included in the analysis for patients with multiple ICU admissions. Exclusion criteria included: 1) hospital stay > 200 days; 2) patients < 18 years old; and 3) elective admissions.

## Demographical and laboratory variables

Demographic data including gender, age, admission type, ethnicity, etiology for CA and ICU type were used for analysis. Physiological variables were extracted for the first and third days after ICU admission. These variables included urine output, use of vasopressors (including norepinephrine, dopamine, epinephrine and dobutamine), and the Glasgow coma score (GCS). Sequential organ failure assessment (SOFA) score on the first day was computed.

Laboratory variables such as creatinine, potassium, total bilirubin, activated partial thrombin time (aPTT), lactate, international normalized ratio (INR), creatinine, sodium, platelet and hematocrit on day 1 and 3 after ICU admission were extracted from the database. The variable associated with the greatest severity of illness was obtained for those with multiple measurements. We also extracted vital signs including heart rate, mean BP, respiratory rate and body temperature on day 1 and 3. The hospital mortality was the primary outcome, which was defined by the vital status at the time of hospital discharge. Missing values were handled by using multiple imputations [26].

## Latent profile and latent transition analysis

The LTA and LPA are closely related methodology. LPA is able to identify latent subgroups of a population by clinical features. LTA extends the methodology of LPA by identifying the movement between the subgroups over time based on longitudinal data. In our study, laboratory tests and vital signs on day 1 and 3 after ICU admission were used to identify the hidden groups. Subphenotypes were identified on day 1 and day 3. The appropriate number of profiles was determined by both clinical relevance and model fit metrics. In this study, the number of profiles were determined by entropy, Bayesian information criteria (BIC), Akaike information criterion (AIC) and Vuong-Lo-Mendell-Rubin Likelihood ratio test (LRT) likelihood ratio tests. Lower values of the AIC and BIC indicates a better model fit [27]. Higher value of entropy indicates higher model fit. A p value less than 0.05 for the LRT was used to judge the superiority of n-profile model to (n-1)-profile model [28]. Furthermore, the patient proportion in each profile should be greater than 5% [29].

The LTA model estimates the latent profile models on day 1 and 3 simultaneously. The relationship of the latent classes between day 1 and 3 was also considered. The LTA model provided an estimate of profile membership on day 1 and 3, as well as the probability of profile transition. All variables used for latent profile model were also included for LTA modeling [18].

## Statistical analysis

Descriptive statistics were performed in standard way. Continuous variables were expressed as median (interquartile range [IQR]) or the mean (standard deviation) as appropriate. Differences between groups were compared using analysis of variance (ANOVA) [21, 22].

An interaction term between profile on day 1 and 3 were included in a multivariable logistic regression model to investigate the effect of each transition pattern on mortality outcome. Potential confounders including age, the SOFA score, ethnicity, ICU type, mean BP and time period of admission (2008–2012 versus before 2008) were adjusted for in the model. All statistical analyses were performed using Mplus (version 7.4) and R package (version 3.4.3). A two tailed p-value < 0.05 was considered as statistical significance.

## Results

### Patient selection

A total of 52,963 admissions were initially identified from the MIMIC-III database. Subjects were excluded as per the exclusion criteria: 101 patients were excluded because they were younger than 18 years old; seven admissions were excluded because of LOS >200 days; and 7391 were excluded because of the elective admission. In the remaining patients, 1361 were ICU admissions due to cardiac arrest. Eight admissions were not the first ICU admission and were further excluded. A number of 504 patients were excluded because they stayed in ICU for less than 3 days. As a result, a number of 848 CA subjects were included for the final analysis (Fig 1).

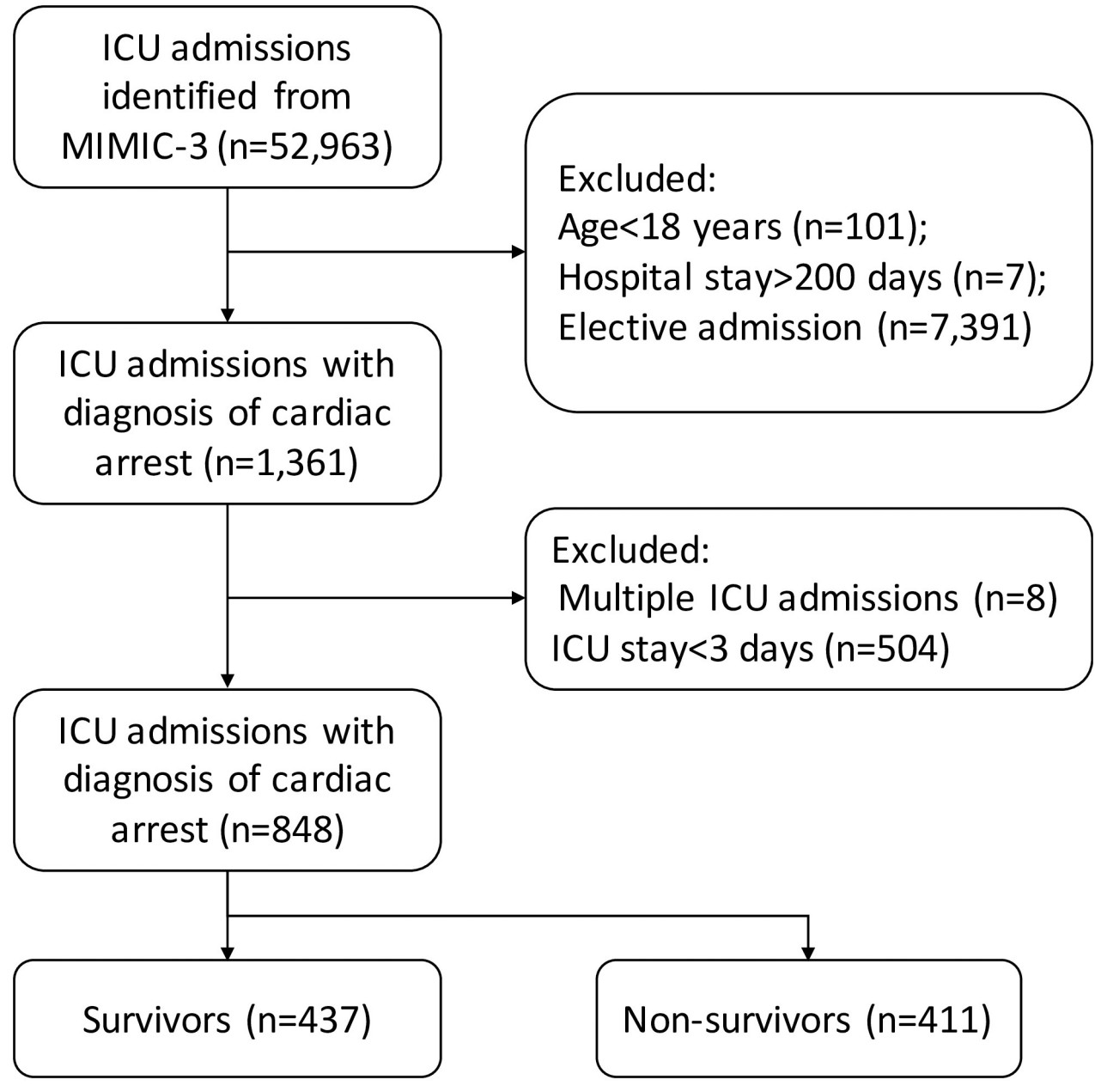

**Fig 1. Flowchart identification of eligible patients from the database.**

**Table 1. Choose the number of profiles on day 1.**

| Number of profiles | BIC | LL | AIC | aBIC | Entropy | AICC | P* | Number of subjects in each profile (%) | | | | | | |
|---|---|---|---|---|---|---|---|---|---|---|---|---|---|---|
| | | | | | | | | 1 | 2 | 3 | 4 | 5 | 6 | 7 |
| 2 | 119900.6 | -59724.40 | 119582.8 | 119687.8 | 1.000 | 119594.5 | 0.2860 | 17(2) | 831(98) | | | | | |
| 3 | 118928.9 | -59161.00 | 118502.0 | 118643.1 | 0.962 | 118523.6 | 0.0464 | 621(73) | 112(13) | 115(14) | | | | |
| 4 | 118561.6 | -58899.81 | 118025.6 | 118202.7 | 0.956 | 118060.7 | 0.3778 | 113(13) | 34(4) | 696(82) | 5(1) | | | |
| 5 | 117663.4 | -58373.17 | 117018.3 | 117231.5 | 0.947 | 117070.8 | 0.3369 | 113(13) | 29(3) | 596(70) | 105(12) | 5(1) | | |
| 6 | 117317.2 | -58122.55 | 116563.1 | 116812.3 | 0.954 | 116637.0 | 0.3234 | 17(2) | 115(14) | 652(77) | 5(1) | 26(3) | 33(4) | |
| 7 | 117166.5 | -57969.67 | 116303.3 | 116588.6 | 0.944 | 116403.5 | 0.3152 | 26(3) | 597(70) | 17(2) | 113(13) | 33(4) | 57(7) | 5(1) |

*The n-profile model was compared to the (n-1)-profile model and p value was reported based on the VUONG-LO-MENDELL-RUBIN likelihood ratio test.

Abbreviations: AICC: Akaike Information Criterion corrected; AIC: Akaike Information Criterion; aBIC: adjusted Bayesian information criteria; BIC: Bayesian information criteria.

## The best number of latent profiles

LPA models with varying number of profiles were fitted and compared for their model fit. The AIC and BIC values dropped rapidly from 2 to 3-profile model (dropped by 1000). The entropy dropped remarkably from a 3-profile model to 4-profile model (from 0.962 to 0.956). The LRT did not show a significant model fit gain for the 3 and 4-profile models. Furthermore, the profile 2 and 4 contained less than 5% of patients in the 4-profile model. Collectively, the 3-profile model was selected the model with best fit metrics (Table 1).

On day 3, the 3-profile model was also chosen as the best model by considering all metrics (Table 2). The LRT showed that the 3-profile model was not significantly worse than the 4-profile model and entropy also indicated better model fit for 3-profile model than 4-profile model. Profile 4 in the 4-profile model contained less than 5% of the overall population (4%).

## Clinical features of the CA profiles

On day 1, *Profile 1* accounted for the largest proportion (73%) and could be considered as the baseline subphenotype. *Profile 2* (13%) was characterized by neurological injury (with a low GCS). The hallmark features of *Profile 3* (14%) was multiple organ dysfunctions involving hepatic injury (high bilirubin), coagulopathy (decreased platelet count, prolonged aPTT and

**Table 2. Statistical metrics for determining the best number of profiles on day 3.**

| Number of profiles | LL | AIC | BIC | aBIC | Entropy | AICC | P* | Number of subjects in each profile (%) | | | | | | |
|---|---|---|---|---|---|---|---|---|---|---|---|---|---|---|
| | | | | | | | | 1 | 2 | 3 | 4 | 5 | 6 | 7 |
| 2 | -56606.75 | 113347.5 | 113665.3 | 113452.5 | 0.996 | 113359.2 | 0.0162 | 806(95) | 42(5) | | | | | |
| 3 | -56103.89 | 112387.8 | 112814.6 | 112528.8 | 0.990 | 112409.4 | 0.0000 | 704(83) | 104(12) | 39(5) | | | | |
| 4 | -55811.90 | 111849.8 | 112385.8 | 112026.9 | 0.959 | 111884.9 | 0.5678 | 103(12) | 602(71) | 108(13) | 35(4) | | | |
| 5 | -55490.60 | 111253.2 | 111898.2 | 111466.3 | 0.971 | 111305.6 | 0.1982 | 100(12) | 85(10) | 609(72) | 21(2) | 33(4) | | |
| 6 | -55426.25 | 111170.5 | 111924.6 | 111419.7 | 0.853 | 111244.5 | 0.7622 | 24(3) | 105(12) | 396(47) | 272(32) | 32(4) | 19(2) | |
| 7 | -55139.43 | 110642.9 | 111506.1 | 110928.1 | 0.977 | 110743.0 | 0.8138 | 7(1) | 32(4) | 85(10) | 596(70) | 20(2) | 95(11) | 12(1) |

*P value was obtained by comparing the n-profile model to the (n-1)-profile model according to the VUONG-LO-MENDELL-RUBIN likelihood ratio test.

Abbreviations: AICC: Akaike Information Criterion corrected; AIC: Akaike Information Criterion; aBIC: adjusted Bayesian information criteria; BIC: Bayesian information criteria.

INR), renal injury (elevated creatinine and low urine output), lung injury (low SPO2) and circulatory failure (with elevated serum lactate and low BP).

*Profile 1* was more likely to be from the coronary care unit (CCU) (34%) and *profile 3* was more likely to be from Trauma Surgical ICU (TSICU) (15%). Patients in *profile 3* were more likely to use vasopressors and inotropes (<0.05). The minimum GCS was significantly lower than other profiles (median: 3; IQR: 3 to 7; p < 0.001). Blood pressure was significantly lower in *profile 3* than in other profiles (p<0.001). There was no statistically significant difference in mortality outcome among profiles on day 1 (Table 3).

### Interventions for the three profiles of CA

There were significant differences in the drug intervention across the three subtypes of CA. For example, profile 3 was more likely to use dopamine than profile 1 and 2. While profile 2 used more norepinephrine than profile 1, profile 1 used more dopamine than profile 2 (Table 3). Profile 3 was more likely to use circulatory support than profile 2 (19% vs. 5%; p = 0.008).

### Latent transition analysis

LTA showed that three profiles could be identified at day 1 and day 3, and the results were very similar to that obtained from LPA. The entropy was 0.970. Characteristics of profiles on day 1 and 3 are shown in Fig 2. Consistent with the LPA analysis, profile 1 was the baseline subphenotype; profile 2 was characterized by neurological injury and profile 3 was characterized by multiple organ dysfunctions.

As shown in Table 4, 535 (85%) patients in profile 1 on day 1 remained in the same profile on day 3. Sixty-one patients (10%) transitioned to *profile 2* and 36 (6%) transitioned to *profile 3*, indicating deterioration of the disease. A majority of patients in *profile 2* (72%) and 3 (82%) on day 1 switched to profile 1 on day 3, indicating improvement in medical condition after treatment (Table 4).

### Impact of therapeutic intervention on profile transition

The associations of medical interventions, such as vasoactive agents and circulatory support, with the transition pattern were explored by univariate analysis. We compared patients transitioned from profile 2 and 3 to profile 1 versus those did not transition to profile 1 (Table 5). The results showed that there was no significant difference between the two transition groups in terms of medications and mechanical circulatory support. Probably, the transition pattern is the intrinsic nature of the disease progression, and current study is not able to identify effective therapeutic intervention that could change the transition path.

### Clinical outcomes of CA profiles based on LTA

There was no statistical difference in hospital mortality between the three latent profiles on day 1 after adjusting for other covariates (Table 6). In the logistic regression model incorporating interaction between profiles on day 1 and 3 (Table 7), the result showed that patients in *profile 1* on day 1 transitioned to *profile 3* had worse survival outcome than those remained in *profile 1* (OR: 20.64; 95% CI: 6.01 to 70.94; p<0.001) and transitioned to *profile 2* (OR: 8.42; 95% CI: 2.22 to 31.97; p = 0.002) on day 3. Patients remained in *profile 1* showed significantly better outcome than those transitioned to *profile 2* (OR: 0.41; 95% CI: 0.23 to 0.73; p = 0.002). For patients in *profile 2* on day 1, patients transitioned to *profile 1* showed significantly better outcome than those remained *profile 2* (OR: 0.30; 95% CI: 0.12 to 0.78; p = 0.014). For patients

**Table 3. Baseline clinical characteristics and outcomes stratified by profiles on the first day.**

| Characteristics | Total (n = 848) | Profile 1 (n = 632) | Profile 2 (n = 116) | Profile 3 (n = 100) | p |
|---|---|---|---|---|---|
| Age, years (IQR) | 67.96(56.96,79.28) | 68.64(58.59,79.76) | 68.1(55.45,79.72) | 64.62(51.88,76.19) | 0.029 |
| Gender, Male (%) | 524(62) | 388(61) | 77(66) | 59(59) | 0.495 |
| Etiology, n (%) | | | | | 0.008 |
| ARF | 257 (30) | 175 (28) | 50 (43) | 32 (32) | |
| MI | 129 (15) | 99 (16) | 15 (13) | 15 (15) | |
| Others | 293 (35) | 234 (37) | 30 (26) | 29 (29) | |
| Sepsis | 144 (17) | 109 (17) | 19 (16) | 16 (16) | |
| Trauma | 25 (3) | 15 (2) | 2 (2) | 8 (8) | |
| Ethnicity, n(%) | | | | | 0.030 |
| ASIAN | 16(2) | 10(2) | 6(5) | 0(0) | |
| BLACK | 69(8) | 43(7) | 12(10) | 14(14) | |
| HISPANIC | 30(4) | 23(4) | 5(4) | 2(2) | |
| UNKNOWN | 132(16) | 104(16) | 17(15) | 11(11) | |
| WHITE | 601(71) | 452(72) | 76(66) | 73(73) | |
| Admission period, n (%) | | | | | < 0.001 |
| Before 2008 | 489(58) | 384(61) | 39(34) | 66(66) | |
| 2008 to 2012 | 359(43) | 248(40) | 77(66) | 34(34) | |
| GCS, median (IQR) | 15(14,15) | 15(15,15) | 3(3,7) | 15(15,15) | < 0.001 |
| SOFA, median (IQR) | 6(4,9) | 5(3,8) | 9.5(8,12) | 10(7,12) | < 0.001 |
| Mean MBP, median (IQR) | 55(47,62) | 55.83(48,62) | 56(47,63) | 48(32,56) | < 0.001 |
| Minimum MBP, median (IQR) | 76.89(70.05,83.88) | 76.64(69.61,83.58) | 80.38(72.85,87.88) | 74.97(70.19,82.45) | 0.002 |
| Care unit type, n (%) | | | | | 0.001 |
| CCU | 265(31) | 212(34) | 35(30) | 18(18) | |
| CSRU | 145(17) | 102(16) | 17(15) | 26(26) | |
| MICU | 270(32) | 193(31) | 49(42) | 28(28) | |
| SICU | 87(10) | 63(10) | 11(9) | 13(13) | |
| TSICU | 81(10) | 62(10) | 4(3) | 15(15) | |
| Use of vasoactive agents | | | | | |
| Dopamine, n (%) | 161(19) | 114(18) | 19(16) | 28(28) | 0.046 |
| Epinephrine, n (%) | 79(9) | 48(8) | 9(8) | 22(22) | < 0.001 |
| Norepinephrine, n (%) | 272(32) | 166(26) | 40(34) | 66(66) | < 0.001 |
| Dobutamine, n (%) | 37(4) | 28(4) | 1(1) | 8(8) | 0.037 |
| MCS/ECMO, n (%) | 102 (12) | 77 (12) | 6 (5) | 19 (19) | 0.008 |
| Clinical outcomes | | | | | |
| Hospital LOS, days (IQR) | 12(7,22) | 12(7,21) | 12(7,24) | 12.5(6.75,22) | 0.844 |
| ICU LOS, days (IQR) | 7(4,13) | 7(4,12) | 7(4,14) | 8(4,15) | 0.386 |
| Hospital mortality, n (%) | 411(48) | 298(47) | 60(52) | 53(53) | 0.416 |

Abbreviations: ICU: intensive care unit; LOS: length of stay; UO: urine output; GCS: Glasgow coma scale; CCU: coronary artery unit; BP: blood pressure; SOFA: sequential organ failure assessment; CSRU: cardiac surgery recovery unit; SICU: surgical ICU; MICU: medical ICU; TSICU: Trauma-Neuro ICU; ECMO: Extracorporeal membrane oxygenation; ARF: acute respiratory failure; MI: myocardial infarction; MCS: mechanical circulatory support.

in *profile 3* on day 1, the transition patterns did not have significant impact on hospital mortality (Table 7).

## Discussion

This study identified three profiles of CA patients based on a large critical care database, and the three profiles were consistent on day 1 and day 3 by using LPA. Consistently, the LTA also

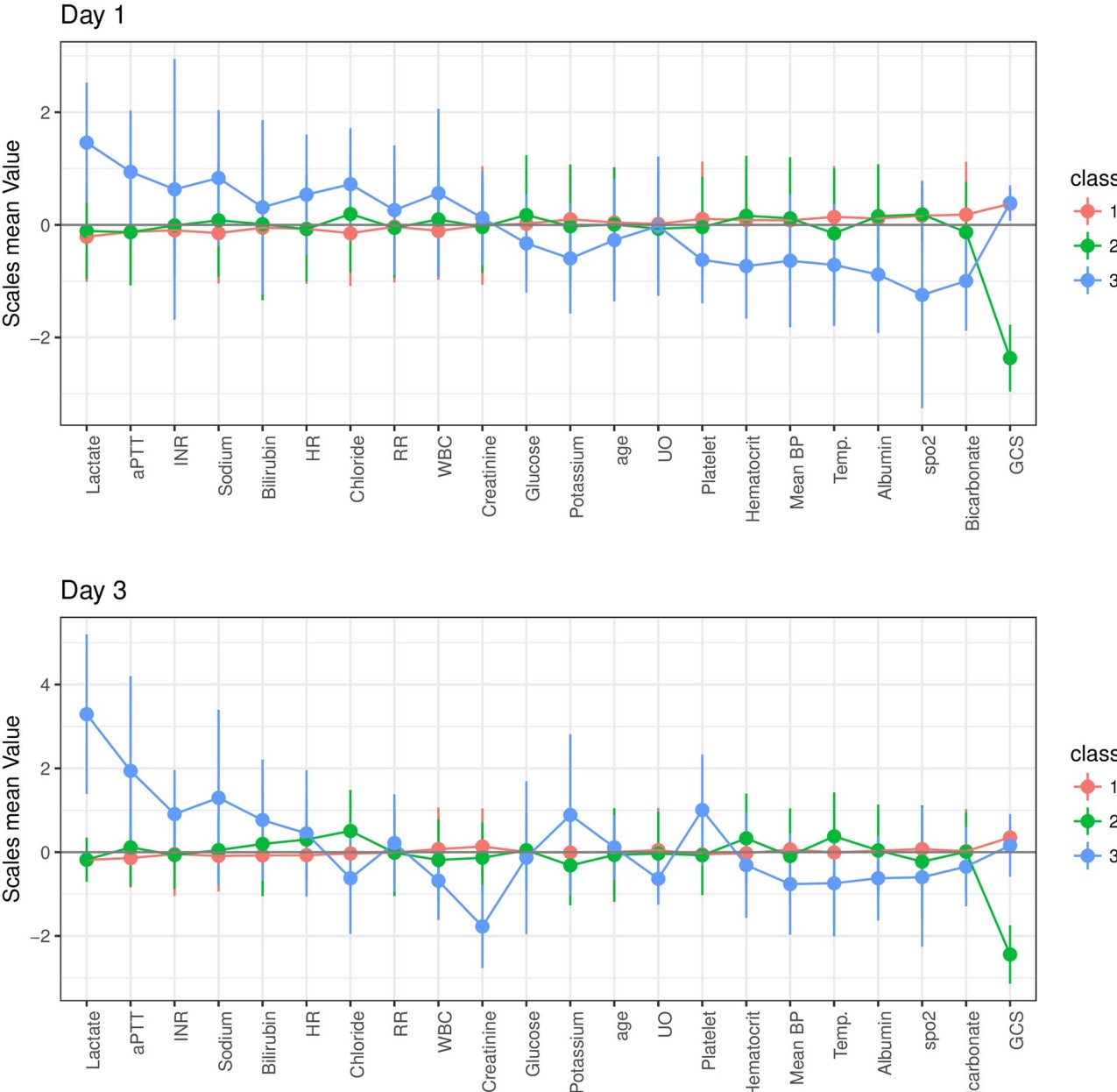

**Fig 2. Clinical characteristics of the three latent profiles on day 1 and 3.** Z-score was normalized by subtracting each individual value by the population mean and divided by the standard deviation. The horizontal line displays the clinical characteristics. Abbreviations: INR: international normalized ratio; aPTT: activated partial thrombin time; HR: heart rate; WBC: white blood cell count; RR: respiratory rate; UO: urine output; BP: blood pressure; GCS: Glasgow Coma Scale.

confirmed the existence of the three subphenotypes. The three subphenotypes were: *Profile 1* (73%) was characterized by the largest proportion of all CA patients and could be considered as the baseline subphenotype; *Profile 2* (13%) was characterized by brain injury with a low GCS; and *Profile 3* (14%) was featured by multiple organ dysfunctions. The same three profiles were identified on day 3 and there were transitions among these profiles. A substantial number of patients in profile 2 (72%) and 3 (82%) on day 1 transitioned to profile 1, suggesting that many patients recovered from multiple organ dysfunction and neurological injury from acute

**Table 4. Transition probability from day 1 to day 3.**

| Day 1 profiles, n (%) | Day 3 profiles, n (%) | | |
|---|---|---|---|
| | 1 | 2 | 3 |
| 1 | 535(85) | 61(10) | 36(6) |
| 2 | 83(72) | 30(26) | 3(3) |
| 3 | 82(82) | 12(12) | 6(6) |

Note: the rows represent the profiles on day 1 after ICU admission, and the column indicates the profiles on day 3. The table shows the number of patients (n, %) transitioned on day 1 and 3.

phase after treatment. Most patients in profile 1 remained assigned to *profile 1* on day 3 (85%), but a minority, 10% and 6% showed deterioration that transitioned to *profile 2* and 3, respectively. Not surprisingly, the mortality outcome was significantly better for patients transitioned to *profile 1* on day 3 than those transitioned or remained in *profile 2* or 3. The profiles of CA identified by the big data analytics are consistent with the pathology of post-cardiac arrest syndrome. The novelty of the study was that we further quantitatively described the epidemiology of the profiles as well as the transitions among these profiles based on a large critical care database. We also quantitatively showed that patients with different transition trajectories presented different clinical outcomes. The transition pattern of post-cardiac arrest syndrome is helpful for risk stratification, which is important for medical resource allocation and decision making. Furthermore, transition pattern might be indicative of medical interventions that can help to direct treatment, although current study failed to identify any difference in interventions across different transition patterns.

An interesting finding in the study was that patients with multiple organ dysfunction and those with neurological injury were categorized into different subgroups. This finding indicates that these patients should be managed differently. Also, the results have implications for the design of clinical trials. For instance, clinical trials designed to investigate agents or

**Table 5. Comparisons between patients who transitioned from profile 2 or 3 to profile 1 versus those not transitioned to profile 1.**

| Variables | Total (n = 216) | Not transition to profile 1 (n = 51) | Transition to profile 1 (n = 165) | p |
|---|---|---|---|---|
| Age (years), Median (IQR) | 66.00 (54.12, 78.10) | 66.33 (55.26, 75.85) | 65.44 (54.07, 78.97) | 0.806 |
| Gender, Male (%) | 136 (63) | 33 (65) | 103 (62) | 0.897 |
| SOFA, Median (IQR) | 10.00 (7.00, 12.00) | 9.00 (7.00, 10.50) | 10.00 (8.00, 12.00) | 0.118 |
| Dopamine use, n (%) | 47 (22) | 7 (14) | 40 (24) | 0.162 |
| Epinephrine use, n (%) | 31 (14) | 7 (14) | 24 (15) | 1.000 |
| Norepinephrine use, n (%) | 106 (49) | 21 (41) | 85 (52) | 0.258 |
| Dobutamine use, n (%) | 9 (4) | 1 (2) | 8 (5) | 0.689 |
| MCS/ECMO, n (%) | 191 (88) | 45 (88) | 146 (88) | 1.000 |
| Etiology, n (%) | | | | 0.777 |
| ARF | 82 (38) | 20 (39) | 62 (38) | |
| MI | 30 (14) | 6 (12) | 24 (15) | |
| Others | 59 (27) | 12 (24) | 47 (28) | |
| Sepsis | 35 (16) | 11 (22) | 24 (15) | |
| Trauma | 10 (5) | 2 (4) | 8 (5) | |

Abbreviations: SOFA: sequential organ failure assessment; ECMO: Extracorporeal membrane oxygenation; MCS: mechanical circulatory support; ARF: acute respiratory failure; MI: myocardial infarction.

**Table 6. Risk factors for hospital mortality on day 1.**

| Features | Odds Ratio | Lower limit of 95% CI | Upper limit of 95% CI | P value |
|---|---|---|---|---|
| Age, with each 10-year increase | 1.04 | 1.01 | 1.07 | 0.014 |
| SOFA (with 1-point increase) | 1.07 | 1.02 | 1.12 | 0.006 |
| Ethnicity (Asia as reference) | | | | |
| BLACK | 1.16 | 0.36 | 3.75 | 0.800 |
| HISPANIC | 0.48 | 0.12 | 1.78 | 0.269 |
| WHITE | 1.03 | 0.35 | 3.04 | 0.952 |
| UNKNOWN | 1.25 | 0.41 | 3.85 | 0.695 |
| Profile 1 as reference | | | | |
| Profile 2 | 0.91 | 0.57 | 1.45 | 0.686 |
| Profile 3 | 1.08 | 0.67 | 1.75 | 0.752 |
| Mean MBP (with each 20-mmHg increase) | 1.07 | 0.81 | 1.42 | 0.616 |
| Care unit type (CCU as reference) | | | | |
| TSICU | 2.17 | 1.29 | 3.67 | 0.004 |
| CSRU | 0.53 | 0.34 | 0.82 | 0.005 |
| MICU | 2.19 | 1.53 | 3.13 | < 0.001 |
| SICU | 1.31 | 0.79 | 2.16 | 0.290 |
| Admission period (before 2008 as reference) | 0.79 | 0.59 | 1.07 | 0.126 |

Abbreviations: CCU: coronary artery unit; SOFA: sequential organ failure assessment; CSRU: cardiac surgery recovery unit; SICU: surgical ICU; MICU: medical ICU; TSICU: Trauma-Neuro ICU.

interventions with neurological protective property should be performed in *profile 2*. In a recent study, Nishikimi M and colleagues showed that the effect of mild therapeutic hypothermia was different depending on the presence or absence of hypoxic encephalopathy [30]. The effects of organ support interventions such as mechanical circulatory support (MCS) and ECMO can be investigated in *profile 3* patients. The study highlighted the importance of individualized treatment for post-resuscitation syndrome. Similarly, trials aiming to investigate management of multiple organ dysfunctions should target patients with profile 3. Profile 3 patients are characterized by circulatory shock and multiple organ failure involving liver, respiratory system, kidney and coagulation. The global ischemia during cardiac arrest leads to rapid release of toxic enzymes and free radicals into circulation, which in turn causes microvascular and metabolic abnormalities of varying degrees [31]. These metabolic disorders were reflected by the deranged laboratory variables associated with multiple organs systems. In our study, the routinely measured laboratory variables and vital signs were used to identify subphenotypes, making our results generalizable to other institutions. Subphenotypes of a disease are usually explored by using genomic information or biomarkers that were not routinely obtained in clinical practice [32, 33]. Although these studies provide more in-depth insights into the underlying pathophysiology of each distinct subphenotype, their clinical utility is limited due to unavailability of these novel biomarkers or transcriptomics. In this regard, the utilization of information collected in the electronic healthcare records was a strength of the present study.

The transition probability between subphenotyes is an interesting finding in our study. It was noted that most patients in *profile 2* (neurological injury) and 3 (multiple organ failure) transitioned to *profile 1* (baseline group) after 3 days treatment. Furthermore, patients transitioned to *profile 1* showed significantly better survival outcome than those remained in *profile 2* or 3. The transition process may reflect treatment strategies used to correct metabolic

**Table 7. Interaction between day 1 and day 3 in multivariable regression model.**

| Variables | Odds Ratio | Lower limit of 95% CI | Upper limit of 95% CI | P value |
|---|---|---|---|---|
| Age, with each 10-year increase | 1.04 | 1.01 | 1.07 | 0.016 |
| SOFA (with 1-point increase) | 1.06 | 1.01 | 1.11 | 0.026 |
| Mean MBP (with each 20-mmHg increase) | 1.03 | 0.77 | 1.38 | 0.843 |
| Ethnicity (Asia as reference) | | | | |
| BLACK | 1.54 | 0.47 | 5.08 | 0.474 |
| HISPANIC | 0.54 | 0.13 | 2.14 | 0.380 |
| WHITE | 1.33 | 0.45 | 3.97 | 0.607 |
| UNKNOWN | 1.65 | 0.53 | 5.19 | 0.387 |
| Care unit type (CCU as reference) | | | | |
| CSRU | 0.41 | 0.25 | 0.66 | < 0.001 |
| SICU | 1.22 | 0.72 | 2.07 | 0.453 |
| TSICU | 2.24 | 1.32 | 3.85 | 0.003 |
| MICU | 2.21 | 1.54 | 3.20 | < 0.001 |
| Admission period (before 2008 as reference) | 0.72 | 0.52 | 0.98 | 0.039 |
| Interaction between day 1 and day 3 profiles | | | | |
| Profile 1 on day 1 | | | | |
| Transition to 3 versus 1 | 20.64 | 6.01 | 70.94 | < 0.001 |
| Transition to 3 versus 2 | 8.42 | 2.22 | 31.97 | 0.002 |
| Stay in 1 versus transition to 2 | 0.41 | 0.23 | 0.73 | 0.002 |
| Profile 2 on day 1 | | | | |
| Transition to 3 versus 1 | 8.39 | 0.7 | 100.33 | 0.093 |
| Transition to 3 versus 2 | 2.54 | 0.19 | 33.52 | 0.480 |
| Transition to 1 versus 2 | 0.3 | 0.12 | 0.78 | 0.014 |
| Profile 3 on day 1 | | | | |
| Stay in 3 versus transition to 1 | 3.4 | 0.37 | 31.21 | 0.279 |
| Stay in 3 versus transition to 2 | 2.36 | 0.19 | 28.84 | 0.501 |
| Transition to 1 versus 2 | 0.69 | 0.19 | 2.6 | 0.588 |

Abbreviations: CCU: coronary artery unit; MICU: medical ICU; SOFA: sequential organ failure assessment; CSRU: cardiac surgery recovery unit; SICU: surgical ICU; TSICU: Trauma-Neuro ICU.

disorders. For example, the goal directed bundle including a target of blood pressure, lactate clearance and urine output was initiated for patients with circulatory shock [34]. In the study, we found that the *profile 3* had significantly greater proportions of the patients on vasopressors and inotropes. There was a substantial number of patients (n = 12; 12%) who transitioned from profile 3 (metabolic disorder) to profile 2 (neurological injury). This indicated the uncontrolled metabolic derangement caused by the ischemia/reperfusion injury would finally lead to brain injury. Neurological outcome is an important component of successful post-resuscitation care. Probability, the brain injury may occur in two stages. Hypoperfusion of the brain directly caused by CA explains brain injury during the first stage, and prolonged metabolic derangement during the post-resuscitation care is responsible for the second stage brain injury.

Several limitations of the study must be acknowledged. First, it is well acknowledged that cardiac arrest may be followed by brain injury, systemic hypo-perfusion, and multiple organ dysfunction. However, there is no direct annotation for the reasons of CA in the database. Thus, we included coexisting diagnosis as possible reasons of the CA. Furthermore, possible reasons for CA can also be deduced by the type of ICU. For example, the possible reason of

CA in CCU is most probably myocardial infarction or arrythmia, and MICU patients may suffer from multiple organ failure. Second, the study failed to find any interventions that were associated with transition patterns. Most probably, the fact that there was no difference in the medications between the patients that transitioned versus those that did not transition could reflect an intrinsic nature of the disease and lack of identifiable interventions that were associated with transitions. Other reasons are that the number of interventions being explored is limited and the study may not have enough power to detect some small effect sizes. Third, our study showed that patients who transitioned from profile 3 to profile 1 did not demonstrate an impact on mortality compared to those who remained in profile 3, which is counterintuitive as judged by clinical expertise. The result showed a trend that patients remained in profile 3 had a 3-fold increase in mortality but the statistical significance was not reached, which could be explained by the small sample size in profile 3 in the study.

In conclusion, our study identified three subphenotypes of CA, which were consistent on day 1 and 3 after ICU admission. There was a substantial transition between these subphenotypes. While most patients experienced recovery after initial therapy especially the ones who transitioned from *profile 2* or 3 to *profile 1*; a minority of patients showed deterioration who transitioned from *profile 1* to *profile 2* or 3. We would like to suggest that clinical trials should be designed by targeting the patient population to specific subphenotypes depending on explored interventions.

## Author Contributions

**Conceptualization:** Lifeng Xing.

**Data curation:** Lifeng Xing.

**Formal analysis:** Lifeng Xing, Min Yao, Zhongheng Zhang.

**Funding acquisition:** Zhongheng Zhang.

**Investigation:** Hemant Goyal.

**Methodology:** Hemant Goyal.

**Project administration:** Yucai Hong.

**Resources:** Min Yao, Yucai Hong.

**Supervision:** Yucai Hong.

**Validation:** Min Yao.

**Visualization:** Hemant Goyal.

**Writing – original draft:** Lifeng Xing.

**Writing – review & editing:** Min Yao, Hemant Goyal, Yucai Hong, Zhongheng Zhang.

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
