## [Decision Letter · Decision Letter 0]

16 Mar 2021

PONE-D-20-40279

Stability of Subphenotypes of Cardiac Arrest Patients Admitted to Intensive Care Unit: a latent transition analysis of a large critical care database

PLOS ONE

Dear Dr. Zhang,

Thank you for submitting your manuscript to PLOS ONE. After careful consideration, we feel that it has merit but does not fully meet PLOS ONE’s publication criteria as it currently stands. Therefore, we invite you to submit a revised version of the manuscript that addresses the points raised during the review process.

We look forward to receiving your revised manuscript.

Kind regards,

Saraschandra Vallabhajosyula, MD MSc

Academic Editor

PLOS ONE

Additional Editor Comments:

I agree with all the comments from the Reviewers. Specifically, Reviewer 3 who mentions overlap with two prior publications - This study with the different subphenotypes has been published earlier, the previous study which has been published in Nature is entitled: Subphenotypes of Cardiac Arrest Patients Admitted to Intensive Care Unit: a latent profile analysis of a large critical care database, link to the article is attached below: https://doi.org/10.1038/s41598-019-50178-0. Most of the information in the above mentioned study has been almost copied verbatim in the present study entitled: Stability of Subphenotypes of Cardiac Arrest Patients Admitted to Intensive Care Unit: a latent transition analysis of a large critical care database. The only additional information in this study in the cross over of subphenotypes and it's effect on mortality on day 3 of hospitalization.

I suggest the authors consider addressing this in great detail to distinguish it from existing papers.

Journal Requirements:

"Z.Z. received funding from The public welfare research project of Zhejiang province

(LGF18H150005) and Scientific research project of Zhejiang Education Commission

(Y201737841);"

5. Thank you for submitting the above manuscript to PLOS ONE. During our internal evaluation of the manuscript, we found significant text overlap between your submission and the following previously published works, some of which you are an author.

https://www.nature.com/articles/s41598-019-50178-0

Please revise the manuscript to rephrase the duplicated text, cite your sources, and provide details as to how the current manuscript advances on previous work. Please note that further consideration is dependent on the submission of a manuscript that addresses these concerns about the overlap in text with published work.

Reviewers' comments:

Reviewer's Responses to Questions

**Comments to the Author**

1. Is the manuscript technically sound, and do the data support the conclusions?

Reviewer #1: Partly

Reviewer #2: Yes

Reviewer #3: Yes

2. Has the statistical analysis been performed appropriately and rigorously? 

Reviewer #1: I Don't Know

Reviewer #2: Yes

Reviewer #3: Yes

3. Have the authors made all data underlying the findings in their manuscript fully available?

Reviewer #1: Yes

Reviewer #2: Yes

Reviewer #3: Yes

4. Is the manuscript presented in an intelligible fashion and written in standard English?

Reviewer #1: Yes

Reviewer #2: Yes

Reviewer #3: Yes

5. Review Comments to the Author

Reviewer #1: The authors of the present study present a well thought out analysis of a secondary database to evaluate sub-phenotypes of cardiac arrest. The authors conclude that the study identified three sub-phenotypes of cardiac arrest which were consistent on day 1 and day 3 after ICU admission. What is unclear is the importance of the study findings. The three profiles identified by the authors (a baseline profile, one with neurologic injury and the third with multiorgan dysfunction) are not new and are recognized to be part of the pathology of post-cardiac arrest syndrome.

It is well acknowledged that cardiac arrest may be followed by brain injury, systemic hypo-perfusion, and multiple organ dysfunction. Therefore, if anything the three profiles identified by the authors indicate patients at varying stages of disease post cardiac arrest. And this sequence of events entirely depends on the timing of arrest, management strategy and importantly underlying pathology.

Can the authors provide any information on the cause of cardiac arrest, timing of arrest along with if and how these patients were managed? Providing analytical results without clinical picture does not add anything to aid in decision making for these patients.

The authors go on to compare outcomes of the three identified profiles. As previously mentioned, it appears these patients are at different stages of the disease and without knowing granular information on the extent and causes of disease, these outcome comparisons are not of value.

It is understandable that sicker patients respond and get better or worse depending on the timeliness, efficacy and other intrinsic factors of treatment strategies. Do we need an analysis to establish this especially with a dynamic condition like post-cardiac arrest pathology?

A more clear presentation of the clinical profiles along with management of these patients may be of value to the reader.

Reviewer #2: I had the chance to review the manuscript “Stability of Subphenotypes of Cardiac Arrest Patients Admitted to Intensive Care Unit: a latent transition analysis of a large critical care database”. The manuscript is well-written, scientifically sound and targets an interesting topic. The authors do a great job in highlighting subgroups that may benefit from individualized management strategies. I have a few comments for the authors’ consideration.

Major comments

1. Inclusion of mechanical circulatory support (MCS) and ECMO as variables for the LTA and LPA would be extremely valuable. It would also be instructive for the readers if the use of MCS and ECMO could be included in Table-3.

2. I would recommend using multiple ICD-9 codes for CA rather than just one code (427.5). ICD-9-CM codes identify CA with varying accuracy and outcomes of CA vary depending on the administrative definition used. (Please refer to Vallabhajosyula et al. Mayo Clin Proc. 2020 doi: 10.1016/j.mayocp.2019.12.007).

3. Include a section on the limitations of the study.

4. If the database allows, stratifying Cardiac arrest by the etiology would enrich the paper. The clinical course, management, and outcomes of CA secondary to different etiologies (ie. MI, Sepsis, ARDS etc ) are quite distinct and targeting individual pathologies would be of benefit.

5. In the discussion the authors mention that the transition process may reflect the treatment strategies used. Would it be possible to assess the different treatment strategies used in patients who transitioned from profile 2 and profile 3 to profile 1 compared to those who remained in their respective profiles?

6. The authors discuss the impact of using the profiling of cardiac arrest patients for clinical trials. It would be informative for readers if they could discuss a proposed clinical approach / management strategy to the 3 distinct profiles encountered on day 1.

Minor comments

1. Consider discussing why the patients who transitioned from profile 3 to profile 1 did not demonstrate an impact on mortality compared to those who remained in profile 3.

2. Line 6 in the Introduction section tends to suggest that TTM has only shown success only in animal studies and not human studies which is not accurate as several studies demonstrated that TTM has improved survival and neurological function in patients with CA. This could be rephrased. (Please refer to the following doi:10.1161/CIR.0000000000000313; doi:10.1056/NEJMoa012689; doi:10.1056/NEJMoa003289).

3. Page 18 Line 23. “In this study, we found that the profile 2 had significantly greater…” Shouldn’t it be profile 3.

4. Line 9 of the discussion should be ‘transitioned to profile 1’ instead of ‘profile 3’.

5. The last sentence of the first paragraph in the discussion section needs to be rephrased.

Reviewer #3: In this original manuscript entitled,” Stability of Subphenotypes of Cardiac Arrest Patients Admitted to Intensive Care Unit:

a latent transition analysis of a large critical care database”,. This is a retrospective US based critical care database study that classifies cardiac arrest into three subphenotypes using the latent transition analysis and evaluates the stability of the three subphenotypes and effects on the ICU mortality outcomes. 848 patients were included and the study demonstrates that patients who transitioned to subphenotype 3 on day 3 of hospitalization had worse survival outcomes.

major comments

1. The introduction and discussion can be more focused on the implications of the stability and transition of subphenotypes

2. This study is similar to your previous study entitled ' Subphenotypes of Cardiac Arrest Patients Admitted to Intensive Care Unit: a latent profile analysis of a large critical care database. would recommend you mention the previous study and highlight the findings of the new study.

6. PLOS authors have the option to publish the peer review history of their article (what does this mean?). If published, this will include your full peer review and any attached files.

Reviewer #1: No

Reviewer #2: **Yes: **Dhiran Verghese

Reviewer #3: **Yes: **Aditi Shankar

---

## [Author Response · Author response to Decision Letter 0]

22 Mar 2021

To Dr. Saraschandra Vallabhajosyula

Academic Editor 

PLOS ONE

Dear Dr. Vallabhajosyula

We thank you and reviewers for the generous comments on the manuscript and have revised the manuscript to address these concerns. Here we enclose our point-by-point responses to the comments raised by the reviewers and editors. We hope our responses and revisions made to the manuscript can address these concerns. We are looking forward to your positive response. 

Yours sincerely,

Zhongheng Zhang, MD

Department of Emergency Medicine, 

Sir Run Run Shaw Hospital, 

Zhejiang University School of Medicine, 

Hangzhou, 

310016, 

China. 

Email: zh_zhang1984@zju.edu.cn

Additional Editor Comments:

REVIEWER COMMENT: I agree with all the comments from the Reviewers. Specifically, Reviewer 3 who mentions overlap with two prior publications - This study with the different subphenotypes has been published earlier, the previous study which has been published in Nature is entitled: Subphenotypes of Cardiac Arrest Patients Admitted to Intensive Care Unit: a latent profile analysis of a large critical care database, link to the article is attached below: https://doi.org/10.1038/s41598-019-50178-0. Most of the information in the above mentioned study has been almost copied verbatim in the present study entitled: Stability of Subphenotypes of Cardiac Arrest Patients Admitted to Intensive Care Unit: a latent transition analysis of a large critical care database. The only additional information in this study in the cross over of subphenotypes and it's effect on mortality on day 3 of hospitalization.

I suggest the authors consider addressing this in great detail to distinguish it from existing papers.

RESPONSE: Many thanks for the constructive suggestions and we thoroughly revised the manuscript to avoid verbatim. Futhermore, we added some lines in the introduction section to describe our previous work and how can the present study add to the existing literature. 

RELATED REVISED MANUSCRIPT TEXT (or Table/Figure): Our previous work has identified subphenotypes of CA using cross-sectional data on the first day of ICU entry [21]. However, it is largely unknown whether the subphenotypes are stable or subject to transitions and how this transition can impact clinical outcomes. Other studies show that subphenotype transition can have significant clinical implications [17,22]. Thus, the current study aimed to characterize the latent transition pattern of CA patients by using latent transition analysis (LTA). The differences in the mortality outcome for patients with different transition paths were also explored. 

MANUSCRIPT LOCATION: P3 L24

Reviewer #1: 

REVIEWER COMMENT: The authors of the present study present a well thought out analysis of a secondary database to evaluate sub-phenotypes of cardiac arrest. The authors conclude that the study identified three sub-phenotypes of cardiac arrest which were consistent on day 1 and day 3 after ICU admission. What is unclear is the importance of the study findings. The three profiles identified by the authors (a baseline profile, one with neurologic injury and the third with multiorgan dysfunction) are not new and are recognized to be part of the pathology of post-cardiac arrest syndrome.

RESPONSE: We are grateful to the reviewer for this insightful comments. We agree with the reviewer for that the subphenotypes identified were consistent with the pathology of post-cardiac arrest syndrome. However, our study utilized big EHR data and analytics to support the pathology of post-cardiac arrest syndrome, making it more evidence based. Furthermore, we showed that these subphenotypes had important clinical implications for prognosis. We revised the manuscript in the discussion section to incorporate the important interpretation provided by the reviewer. 

RELATED REVISED MANUSCRIPT TEXT (or Table/Figure): The profiles of CA identified by the big data analytics are consistent with the pathology of post-cardiac arrest syndrome. The novelty of the study is that we further quantitatively described the epidemiology of the profiles as well as the transitions among these profiles based on a large critical care database. We also quantitatively show that patients with different transition trajectories present different clinical outcomes. 

MANUSCRIPT LOCATION: P17 L15

REVIEWER COMMENT: It is well acknowledged that cardiac arrest may be followed by brain injury, systemic hypo-perfusion, and multiple organ dysfunction. Therefore, if anything the three profiles identified by the authors indicate patients at varying stages of disease post cardiac arrest. And this sequence of events entirely depends on the timing of arrest, management strategy and importantly underlying pathology.

RESPONSE: we fully agree with the reviewer for the constructive comments. We added possible reasons for CA in the revision according to co-existing diagnosis. In the database, there is no direct annotation to the cardiac arrest reasons; thus we also use the information of the type of ICU to deduce possible reasons for CA. we also added this point as a limitation in the revision. 

Some drug interventions were also added to the table to implicate some difference in management between subtypes.

RELATED REVISED MANUSCRIPT TEXT (or Table/Figure): 

Table 3

Variables Total (n=848) Profile 1 (n=632) Profile 2 (n=116) Profile 3 (n=100) p

Etiology, n (%) 0.008

 ARF 257 (30) 175 (28) 50 (43) 32 (32) 

 MI 129 (15) 99 (16) 15 (13) 15 (15) 

 Others 293 (35) 234 (37) 30 (26) 29 (29) 

 Sepsis 144 (17) 109 (17) 19 (16) 16 (16) 

 Trauma 25 ( 3) 15 (2) 2 (2) 8 (8) 

Type of care unit, n(%) 0.001

 CCU 265(31) 212(34) 35(30) 18(18) 

 CSRU 145(17) 102(16) 17(15) 26(26) 

 MICU 270(32) 193(31) 49(42) 28(28) 

 SICU 87(10) 63(10) 11(9) 13(13) 

 TSICU 81(10) 62(10) 4(3) 15(15) 

Dopamine, n(%) 161(19) 114(18) 19(16) 28(28) 0.046

Epinephrine, n(%) 79(9) 48(8) 9(8) 22(22) 0.000

Norepinephrine, n(%) 272(32) 166(26) 40(34) 66(66) 0.000

Dobutamine, n(%) 37(4) 28(4) 1(1) 8(8) 0.037

Abbreviations: CCU: coronary artery unit; CSRU: cardiac surgery recovery unit; MICU: medical ICU; SICU: surgical ICU; TSICU: Trauma-Neuro ICU; ARF: acute respiratory failure; MI: myocardial infarction. 

First, it is well acknowledged that cardiac arrest may be followed by brain injury, systemic hypo-perfusion, and multiple organ dysfunction. However, there is no direct annotation for the reasons of CA in the database. Thus, we included coexisting diagnosis as possible reasons of the CA. Furthermore, possible reasons for CA can also be deduced by the type of ICU. For example, the possible reason of CA in CCU is most probably myocardial infarction or arrythmia, and MICU patients may suffer from multiple organ failure.

MANUSCRIPT LOCATION: Table 3. P19 L11.

REVIEWER COMMENT: Can the authors provide any information on the cause of cardiac arrest, timing of arrest along with if and how these patients were managed? Providing analytical results without clinical picture does not add anything to aid in decision making for these patients.

RESPONSE: we fully agree with the reviewer on this point that analytical results without clinical picture does not add anything to aid in decision making for these patients. However,there is no direct annotation of the reasons for CA events in the database. We added co-existing diagnosis as possible reasons for the CA. We also reported the type of ICU for possible reasons of CA. For example, the possible reason of CA in CCU is most probably myocardial infarction or arrythmia, and MICU patients may suffer from multiple organ failure. We added vasopressors and mechanical circulatory support as the intervention to the results as this information is well recorded in the database. 

RELATED REVISED MANUSCRIPT TEXT (or Table/Figure): 

Table 3

Variables Total (n=848) Profile 1 (n=632) Profile 2 (n=116) Profile 3 (n=100) p

Etiology, n (%) 0.008

 ARF 257 (30) 175 (28) 50 (43) 32 (32) 

 MI 129 (15) 99 (16) 15 (13) 15 (15) 

 Others 293 (35) 234 (37) 30 (26) 29 (29) 

 Sepsis 144 (17) 109 (17) 19 (16) 16 (16) 

 Trauma 25 ( 3) 15 (2) 2 (2) 8 (8) 

Type of care unit, n(%) 0.001

 CCU 265(31) 212(34) 35(30) 18(18) 

 CSRU 145(17) 102(16) 17(15) 26(26) 

 MICU 270(32) 193(31) 49(42) 28(28) 

 SICU 87(10) 63(10) 11(9) 13(13) 

 TSICU 81(10) 62(10) 4(3) 15(15) 

Dopamine, n(%) 161(19) 114(18) 19(16) 28(28) 0.046

Epinephrine, n(%) 79(9) 48(8) 9(8) 22(22) 0.000

Norepinephrine, n(%) 272(32) 166(26) 40(34) 66(66) 0.000

Dobutamine, n(%) 37(4) 28(4) 1(1) 8(8) 0.037

MCS/ECMO, n (%) 102 (12) 77 (12) 6 (5) 19 (19) 0.008

Abbreviations: CCU: coronary artery unit; CSRU: cardiac surgery recovery unit; MICU: medical ICU; SICU: surgical ICU; TSICU: Trauma-Neuro ICU.

First, it is well acknowledged that cardiac arrest may be followed by brain injury, systemic hypo-perfusion, and multiple organ dysfunction. However, there is no direct annotation for the reasons of CA in the database. Thus, we included coexisting diagnosis as possible reasons of the CA. Furthermore, possible reasons for CA can also be deduced by the type of ICU. For example, the possible reason of CA in CCU is most probably myocardial infarction or arrythmia, and MICU patients may suffer from multiple organ failure.

MANUSCRIPT LOCATION: Table 3. P19 L11.

REVIEWER COMMENT: The authors go on to compare outcomes of the three identified profiles. As previously mentioned, it appears these patients are at different stages of the disease and without knowing granular information on the extent and causes of disease, these outcome comparisons are not of value.

RESPONSE: In line with previous comments, we added the type of ICU/etiology of CA and some drug/mechanical support interventions in the result section. A new section with the subtitle “Interventions for the three profiles of CA” was added in this round of revision. 

RELATED REVISED MANUSCRIPT TEXT (or Table/Figure): 

Interventions for the three profiles of CA

There were significant differences in the drug intervention across the three subtypes of CA. For example, profile 3 was more likely to use dopamine than profile 1 and 2. While profile 2 used more norepinephrine than profile 1, profile 1 used more dopamine than profile 2 (Table 3). Profile 3 was more likely to use circulatory support than profile 2 (19% vs. 5%; p = 0.008).

MANUSCRIPT LOCATION: Table 3; P11 L4

REVIEWER COMMENT: It is understandable that sicker patients respond and get better or worse depending on the timeliness, efficacy and other intrinsic factors of treatment strategies. Do we need an analysis to establish this especially with a dynamic condition like post-cardiac arrest pathology?

RESPONSE: We are sorry for not making the rationale of the study clear and straightforward in the first version. We added some lines in the introduction in this round of revision to clarify this point. The rationale for this study is described in the introduction.

RELATED REVISED MANUSCRIPT TEXT (or Table/Figure): Our previous work has identified subphenotypes of CA using cross-sectional data on the first day of ICU entry [21]. However, it is largely unknown whether the subphenotypes are stable or subject to transitions and how this transition can impact clinical outcomes. Other studies show that subphenotype transition can have significant clinical implications [17,22]. Thus, the current study aimed to characterize the latent transition pattern of CA patients by using latent transition analysis (LTA). The differences in the mortality outcome for patients with different transition paths were also explored. 

MANUSCRIPT LOCATION: P3 L24

REVIEWER COMMENT: A more clear presentation of the clinical profiles along with management of these patients may be of value to the reader.

RESPONSE: We fully agree with the reviewer in this point. In the revision, we added some more sections to show the medication and clinical difference between the clinical profiles.

RELATED REVISED MANUSCRIPT TEXT (or Table/Figure): 

Table 3

Variables Total (n=848) Profile 1 (n=632) Profile 2 (n=116) Profile 3 (n=100) p

Type of care unit, n(%) 0.001

 CCU 265(31) 212(34) 35(30) 18(18) 

 CSRU 145(17) 102(16) 17(15) 26(26) 

 MICU 270(32) 193(31) 49(42) 28(28) 

 SICU 87(10) 63(10) 11(9) 13(13) 

 TSICU 81(10) 62(10) 4(3) 15(15) 

Dopamine, n(%) 161(19) 114(18) 19(16) 28(28) 0.046

Epinephrine, n(%) 79(9) 48(8) 9(8) 22(22) 0.000

Norepinephrine, n(%) 272(32) 166(26) 40(34) 66(66) 0.000

Dobutamine, n(%) 37(4) 28(4) 1(1) 8(8) 0.037

MCS/ECMO, n (%) 102 (12) 77 (12) 6 (5) 19 (19) 0.008

Abbreviations: CCU: coronary artery unit; CSRU: cardiac surgery recovery unit; MICU: medical ICU; SICU: surgical ICU; TSICU: Trauma-Neuro ICU; MCS: mechanical circulatory support; ECMO: Extracorporeal membrane oxygenation.

Interventions for the three profiles of CA

There were significant differences in the drug intervention across the three subtypes of CA. For example, profile 3 was more likely to use dopamine than profile 1 and 2. While profile 2 used more norepinephrine than profile 1, profile 1 used more dopamine than profile 2 (Table 3). Profile 3 was more likely to use circulatory support than profile 2 (19% vs. 5%; p = 0.008).

MANUSCRIPT LOCATION: Table 3; P11 L4

Reviewer #2: 

REVIEWER COMMENT: I had the chance to review the manuscript “Stability of Subphenotypes of Cardiac Arrest Patients Admitted to Intensive Care Unit: a latent transition analysis of a large critical care database”. The manuscript is well-written, scientifically sound and targets an interesting topic. The authors do a great job in highlighting subgroups that may benefit from individualized management strategies. I have a few comments for the authors’ consideration.

RESPONSE: Thank you for the insightful comments. We have revised the manuscrupt with point-by-point responses to each question. 

RELATED REVISED MANUSCRIPT TEXT (or Table/Figure): None

MANUSCRIPT LOCATION: None

Major comments

REVIEWER COMMENT: 1. Inclusion of mechanical circulatory support (MCS) and ECMO as variables for the LTA and LPA would be extremely valuable. It would also be instructive for the readers if the use of MCS and ECMO could be included in Table-3.

RESPONSE: we fully agree with the reviewer on this point and added such information in the revision in table 3. However, we do not agree to include interventions into the LPA/LTA models because these models generally uses features/characteristics. 

RELATED REVISED MANUSCRIPT TEXT (or Table/Figure): 

Table 3 Baseline characteristics and outcomes by profiles on day 1

MCS/ECMO, n (%) 102 (12) 77 (12) 6 (5) 19 (19) 0.008

Profile 3 was more likely to use circulatory support than profile 2 (19% vs. 5%; p = 0.008).

MANUSCRIPT LOCATION: Table 3; P11 L8.

REVIEWER COMMENT: 2. I would recommend using multiple ICD-9 codes for CA rather than just one code (427.5). ICD-9-CM codes identify CA with varying accuracy and outcomes of CA vary depending on the administrative definition used. (Please refer to Vallabhajosyula et al. Mayo Clin Proc. 2020 doi: 10.1016/j.mayocp.2019.12.007).

RESPONSE: we are grateful to the reviewer for this insightful comment. We are sorry for not making the inclusion criteria clear in the original version, we searched both ICD-9 code and diagnosis including cardiac arrest, cardiopulmonary resuscitation and ventricular fibrilation. Thus, although the ICD-9 code for CPR and VF were not used, but we have actually included such patients. We added this to the revision to ensure the inlcuded patients represented cardiac arrest target population. 

RELATED REVISED MANUSCRIPT TEXT (or Table/Figure): Subjects with the diagnosis of cardiopulmonary resuscitation (ICD-9 code: 9960 and 9963), cardiac arrest (ICD-9 code: 4275) and ventricular fibrillation (ICD-9 code: 4274) were screened for potential eligibility[25].

MANUSCRIPT LOCATION: P5 L15

REVIEWER COMMENT: 3. Include a section on the limitations of the study.

RESPONSE: we included a section for limitation in this round of revision.

RELATED REVISED MANUSCRIPT TEXT (or Table/Figure): Several limitations of the study must be acknowledged. First, it is well acknowledged that cardiac arrest may be followed by brain injury, systemic hypo-perfusion, and multiple organ dysfunction. However, there is no direct annotation for the reasons of CA in the database. Thus, we included coexisting diagnosis as possible reasons of the CA. Furthermore, possible reasons for CA can also be deduced by the type of ICU. For example, the possible reason of CA in CCU is most probably myocardial infarction or arrythmia, and MICU patients may suffer from multiple organ failure. Second, the study failed to find any interventions that were associated with transition patterns. Most probably, the transition pattern is an intrinsic nature of the disease, and interventions such as vasopressors and circulatory support cannot change the transition pattern. Another reason is that the interventions being explored are limited, numerous interventions are being used during post-CA care. We cannot explore all of them in one study. Third, our study showed that patients who transitioned from profile 3 to profile 1 did not demonstrate an impact on mortality compared to those who remained in profile 3. This is counterintuitive from clinical expertise. The result showed a trend that patients remained in profile 3 had 3-fold increase in mortality but the statistical significance was not reached, which could be due to the small sample size in profile 3 in the study. 

MANUSCRIPT LOCATION: P19 L11

REVIEWER COMMENT: 4. If the database allows, stratifying Cardiac arrest by the etiology would enrich the paper. The clinical course, management, and outcomes of CA secondary to different etiologies (ie. MI, Sepsis, ARDS etc ) are quite distinct and targeting individual pathologies would be of benefit.

RESPONSE: we fully agree with the reviewer on this point and added these etiology into the analysis in this round of revision.

RELATED REVISED MANUSCRIPT TEXT (or Table/Figure): 

Etiology, n (%) 0.008

 ARF 257 (30) 175 (28) 50 (43) 32 (32) 

 MI 129 (15) 99 (16) 15 (13) 15 (15) 

 Others 293 (35) 234 (37) 30 (26) 29 (29) 

 Sepsis 144 (17) 109 (17) 19 (16) 16 (16) 

 Trauma 25 ( 3) 15 (2) 2 (2) 8 (8) 

Abbreviations: ARF: acute respiratory failure; MI: myocardial infarction.

MANUSCRIPT LOCATION: Table 3

REVIEWER COMMENT: 5. In the discussion the authors mention that the transition process may reflect the treatment strategies used. Would it be possible to assess the different treatment strategies used in patients who transitioned from profile 2 and profile 3 to profile 1 compared to those who remained in their respective profiles?

RESPONSE: we are greateful to the reviewer for this valuable comments. In this revision, we performed more analysis on the differences in treatment strategy for patients with different transition pattern. 

RELATED REVISED MANUSCRIPT TEXT (or Table/Figure): 

Impact of therapeutic intervention on profile transition

The associations of medical interventions, such as vasoactive agents and circulatory support, with the transition pattern were explored by univariate analysis. We compared patients transitioned from profile 2 and 3 to profile 1 versus those did not transition to profile 1 (Table 5). The results showed that there was no significant difference between the two transition groups in terms of medications and mechanical circulatory support. Probably, the transition pattern is the intrinsic nature of the disease progression, there is no effective therapeutic intervention that could change the transition path. 

Table 5. Comparisons between patients who transitioned from profile 2 or 3 to profile 1 versus those not transitioned to profile 1. 

Variables Total (n = 216) Not transition to profile 1 (n = 51) Transition to profile 1 (n = 165) p

Age (years), Median (IQR) 66.00 (54.12, 78.10) 66.33 (55.26, 75.85) 65.44 (54.07, 78.97) 0.806

Gender, Male (%) 136 (63) 33 (65) 103 (62) 0.897

SOFA, Median (IQR) 10.00 (7.00, 12.00) 9.00 (7.00, 10.50) 10.00 (8.00, 12.00) 0.118

Dopamine use, n (%) 47 (22) 7 (14) 40 (24) 0.162

Epinephrine use, n (%) 31 (14) 7 (14) 24 (15) 1.000

Norepinephrine use, n (%) 106 (49) 21 (41) 85 (52) 0.258

Dobutamine use, n (%) 9 ( 4) 1 ( 2) 8 ( 5) 0.689

MCS/ECMO, n (%) 191 (88) 45 (88) 146 (88) 1.000

Etiology, n (%) 0.777

 ARF 82 (38) 20 (39) 62 (38) 

 MI 30 (14) 6 (12) 24 (15) 

 Others 59 (27) 12 (24) 47 (28) 

 Sepsis 35 (16) 11 (22) 24 (15) 

 Trauma 10 ( 5) 2 ( 4) 8 ( 5) 

Abbreviations: SOFA: sequential organ failure assessment; MCS: mechanical circulatory support; ECMO: Extracorporeal membrane oxygenation; ARF: acute respiratory failure; MI: myocardial infarction.

MANUSCRIPT LOCATION: P13 L16; Table 5.

REVIEWER COMMENT: 6. The authors discuss the impact of using the profiling of cardiac arrest patients for clinical trials. It would be informative for readers if they could discuss a proposed clinical approach / management strategy to the 3 distinct profiles encountered on day 1.

RESPONSE: We fully agree with the reviewer on this constructive suggestion. We added more analyses in this revision to explore potential beneficial effects of interventions for patients with different transition pattern, however, no intervention was found to be associted with the transition pattern. We acknowledge this limitation in the revision. Futhermore, we added more discussions on the use of different interventions on day 1.

RELATED REVISED MANUSCRIPT TEXT (or Table/Figure): 

For instance, clinical trials designed to investigate agents or interventions with neurological protective property should be performed in profile 2. In a recent study, Nishikimi M and colleagues showed that the effect of mild therapeutic hypothermia was different depending on the presence or absence of hypoxic encephalopathy [30]. The effects of organ support interventions such as mechanical circulatory support (MCS) and ECMO can be investigated in profile 3 patients.

Second, the study failed to find any interventions that were associated with transition patterns. Most probably, the transition pattern is an intrinsic nature of the disease, and interventions such as vasopressors and circulatory support cannot change the transition pattern. Another reason is that the interventions being explored are limited, numerous interventions are being used during post-CA care. We cannot explore all of them in one study. 

MANUSCRIPT LOCATION: P17 L24; P19 L17

Minor comments

REVIEWER COMMENT: 1. Consider discussing why the patients who transitioned from profile 3 to profile 1 did not demonstrate an impact on mortality compared to those who remained in profile 3.

RESPONSE: we added discussion on this point in the revision.

RELATED REVISED MANUSCRIPT TEXT (or Table/Figure): Third, our study showed that patients who transitioned from profile 3 to profile 1 did not demonstrate an impact on mortality compared to those who remained in profile 3, which is counterintuitive as judged by clinical expertise. The result showed a trend that patients remained in profile 3 had 3-fold increase in mortality but the statistical significance was not reached, which could be explained by the small sample size in profile 3 in the study.

MANUSCRIPT LOCATION: P19 L21

REVIEWER COMMENT: 2. Line 6 in the Introduction section tends to suggest that TTM has only shown success only in animal studies and not human studies which is not accurate as several studies demonstrated that TTM has improved survival and neurological function in patients with CA. This could be rephrased. (Please refer to the following doi:10.1161/CIR.0000000000000313; doi:10.1056/NEJMoa012689; doi:10.1056/NEJMoa003289).

RESPONSE: we rephrased this text in the revision. 

RELATED REVISED MANUSCRIPT TEXT (or Table/Figure): In particular, targeted temperature management (TTM) has been shown to improve survival and neurological functions in patients with CA[9-11].

MANUSCRIPT LOCATION: P3 L8

REVIEWER COMMENT: 3. Page 18 Line 23. “In this study, we found that the profile 2 had significantly greater…” Shouldn’t it be profile 3.

RESPONSE: We are sorry for this mistake and we correct it in the revision. 

RELATED REVISED MANUSCRIPT TEXT (or Table/Figure): In the study, we found that the profile 3 had significantly greater proportions of the patients on vasopressors and inotropes.

MANUSCRIPT LOCATION: P19 L2

REVIEWER COMMENT: 4. Line 9 of the discussion should be ‘transitioned to profile 1’ instead of ‘profile 3’.

RESPONSE: corrected

RELATED REVISED MANUSCRIPT TEXT (or Table/Figure): A substantial number of patients in profile 2 (72%) and 3 (82%) on day 1 transitioned to profile 1

MANUSCRIPT LOCATION: P17 L9

REVIEWER COMMENT: 5. The last sentence of the first paragraph in the discussion section needs to be rephrased.

RESPONSE: corrected

RELATED REVISED MANUSCRIPT TEXT (or Table/Figure): Not surprisingly, the mortality outcome was significantly better for patients transitioned to profile 1 on day 3 than those transitioned or remained in profile 2 or 3.

MANUSCRIPT LOCATION: P17 L13

Reviewer #3: 

REVIEWER COMMENT: In this original manuscript entitled,” Stability of Subphenotypes of Cardiac Arrest Patients Admitted to Intensive Care Unit:

a latent transition analysis of a large critical care database”,. This is a retrospective US based critical care database study that classifies cardiac arrest into three subphenotypes using the latent transition analysis and evaluates the stability of the three subphenotypes and effects on the ICU mortality outcomes. 848 patients were included and the study demonstrates that patients who transitioned to subphenotype 3 on day 3 of hospitalization had worse survival outcomes.

RESPONSE: Thank you for the comments.

RELATED REVISED MANUSCRIPT TEXT (or Table/Figure): None 

MANUSCRIPT LOCATION: None

major comments

REVIEWER COMMENT: 1. The introduction and discussion can be more focused on the implications of the stability and transition of subphenotypes

RESPONSE: We revised the introduction to meet the requirement. 

RELATED REVISED MANUSCRIPT TEXT (or Table/Figure): In addition, the subphenotype transition has also been widely investigated because unraveling the transition pattern can have significant clinical and research implications[16-18]. For example, subphenotype stability over time can help to design trials and/or therapeutics. Subphenotype transition is also important to the question on whether difference in clinical presentation is dependent on the timing of measurement[16]. 

However, it is largely unknown whether the subphenotypes are stable or subject to transitions and how this transition can impact clinical outcomes. Other studies show that subphenotype transition can have significant clinical implications [15,16]. Thus, the current study aimed to characterize the latent transition pattern of CA patients by using latent transition analysis (LTA).

MANUSCRIPT LOCATION: P3 L17; P3 L25

REVIEWER COMMENT: 2. This study is similar to your previous study entitled ' Subphenotypes of Cardiac Arrest Patients Admitted to Intensive Care Unit: a latent profile analysis of a large critical care database. would recommend you mention the previous study and highlight the findings of the new study.

RESPONSE: we cited our previous work in the revision and added some sentences on how can the current study add new. 

RELATED REVISED MANUSCRIPT TEXT (or Table/Figure): Our previous work has identified subphenotypes of CA using cross-sectional data on the first day of ICU entry [14]. However, it is largely unknown whether the subphenotypes are stable or subject to transitions and how this transition can impact clinical outcomes. Other studies show that subphenotype transition can have significant clinical implications [15,16]. Thus, the current study aimed to characterize the latent transition pattern of CA patients by using latent transition analysis (LTA). The differences in the mortality outcome for patients with different transition paths were also explored. 

MANUSCRIPT LOCATION: P3 L24

---

## [Decision Letter · Decision Letter 1]

15 Apr 2021

PONE-D-20-40279R1

Latent transition analysis of cardiac arrest patients treated in the intensive care unit

PLOS ONE

Dear Dr. Zhang,

Thank you for submitting your manuscript to PLOS ONE. After careful consideration, we feel that it has merit but does not fully meet PLOS ONE’s publication criteria as it currently stands. Therefore, we invite you to submit a revised version of the manuscript that addresses the points raised during the review process.

We look forward to receiving your revised manuscript.

Kind regards,

Saraschandra Vallabhajosyula, MD MSc

Academic Editor

PLOS ONE

Journal Requirements:

Reviewers' comments:

Reviewer's Responses to Questions

**Comments to the Author**

1. If the authors have adequately addressed your comments raised in a previous round of review and you feel that this manuscript is now acceptable for publication, you may indicate that here to bypass the “Comments to the Author” section, enter your conflict of interest statement in the “Confidential to Editor” section, and submit your "Accept" recommendation.

Reviewer #2: (No Response)

Reviewer #3: All comments have been addressed

2. Is the manuscript technically sound, and do the data support the conclusions?

Reviewer #2: Yes

Reviewer #3: Yes

3. Has the statistical analysis been performed appropriately and rigorously? 

Reviewer #2: Yes

Reviewer #3: Yes

4. Have the authors made all data underlying the findings in their manuscript fully available?

Reviewer #2: Yes

Reviewer #3: Yes

5. Is the manuscript presented in an intelligible fashion and written in standard English?

Reviewer #2: Yes

Reviewer #3: Yes

6. Review Comments to the Author

Reviewer #2: This paper titled “Latent transition analysis of cardiac arrest patients treated in the intensive care unit” focuses on sub-phenotypes of cardiac arrest patients on day 1 and day 3 of admission and the transition between the identified groups. The paper is scientifically sound and well written. I appreciate the efforts the authors made to address all the concerns raised by the reviewers. I would like to mention a few minor comments I have

1. The manuscript would benefit from copyediting.

2. Although the overlap with your prior manuscript has been addressed, attempts to minimize this further would be appreciated.

3. Rephrase Page 19 line 19, 20 and page 14 line 5 – The fact that there was no difference in the medications between the patients that transitioned vs those that did not transition could reflect an intrinsic nature of the disease and lack of identifiable interventions that were associated with transition. However, concluding that the interventions cannot change transition might not be appropriate as patients requiring vasopressors/ inotrope/ TTM/ circulatory support and not receiving it will surely lead to deleterious effects.

Reviewer #3: I had the opportunity to re-review the manuscript entitled " Latent transition analysis of cardiac arrest patients treated in the intensive care unit".

1.Most of the revisions that have been suggested by me and the other reviewer's and editor has been incorporated in the manuscript.

2. The manuscript has been revised to not mirror it's previous study verbatim entitled ' Subphenotypes of Cardiac Arrest Patients Admitted to Intensive Care Unit: a latent profile analysis of a large critical care database'.

3. The sub-phenotypes of CA indicate a continuum of the disease process of CA and post cardiac arrest syndrome, the manuscript is not clearly demonstrating the clinical implications (not the further research implications) of the sub-phenotypes. The future research implication of the transition of the sub-phenotypes have been discussed in detail in this revision of the manuscript. Furthermore, There is a statement that is contradicting the conclusions and the clinical implications aspect of the discussion. The authors themselves state that ' Probably, the transition pattern is the intrinsic nature of the disease progression, there is no effective therapeutic intervention that could change the transition path' . but further go on to discuss the possible interventions that could be carried out at different sub-phenotype profile. This statement has to be modified or addressed in the discussion as it is contradicting the study aim.

7. PLOS authors have the option to publish the peer review history of their article (what does this mean?). If published, this will include your full peer review and any attached files.

Reviewer #2: **Yes: **Dhiran Verghese

Reviewer #3: **Yes: **Aditi Shankar

---

## [Author Response · Author response to Decision Letter 1]

30 Apr 2021

To Dr. Saraschandra Vallabhajosyula

Academic Editor 

PLOS ONE

Dear Dr. Vallabhajosyula

We thank you and reviewers for the generous comments on the manuscript and have revised the manuscript to address these concerns. Here we enclose our point-by-point responses to the comments raised by the reviewers and editors. We hope our responses and revisions made to the manuscript can address these concerns. We are looking forward to your positive response. 

Yours sincerely,

Zhongheng Zhang, MD

Department of Emergency Medicine, 

Sir Run Run Shaw Hospital, 

Zhejiang University School of Medicine, 

Hangzhou, 

310016, 

China. 

Email: zh_zhang1984@zju.edu.cn

Reviewer #2: This paper titled “Latent transition analysis of cardiac arrest patients treated in the intensive care unit” focuses on sub-phenotypes of cardiac arrest patients on day 1 and day 3 of admission and the transition between the identified groups. The paper is scientifically sound and well written. I appreciate the efforts the authors made to address all the concerns raised by the reviewers. I would like to mention a few minor comments I have

REVIEWER COMMENT: 1. The manuscript would benefit from copyediting.

RESPONSE: the tables in the main text were edited and some other places were also editted for clarity and typos.

RELATED REVISED MANUSCRIPT TEXT (or Table/Figure): multiple places

MANUSCRIPT LOCATION: multiple places.

REVIEWER COMMENT: 2. Although the overlap with your prior manuscript has been addressed, attempts to minimize this further would be appreciated.

RESPONSE: the manuscript was revised futher to reduce overlap.

RELATED REVISED MANUSCRIPT TEXT (or Table/Figure): multiple places

MANUSCRIPT LOCATION: multiple places.

REVIEWER COMMENT: 3. Rephrase Page 19 line 19, 20 and page 14 line 5 – The fact that there was no difference in the medications between the patients that transitioned vs those that did not transition could reflect an intrinsic nature of the disease and lack of identifiable interventions that were associated with transition. However, concluding that the interventions cannot change transition might not be appropriate as patients requiring vasopressors/ inotrope/ TTM/ circulatory support and not receiving it will surely lead to deleterious effects.

RESPONSE: we fully agree with the reviewer on this point and revised the texts.

RELATED REVISED MANUSCRIPT TEXT (or Table/Figure): Second, the study failed to find any interventions that were associated with transition patterns. Most probably, the fact that there was no difference in the medications between the patients that transitioned versus those that did not transition could reflect an intrinsic nature of the disease and lack of identifiable interventions that were associated with transitions. Other reasons are that the number of interventions being explored is limited and the study may not have enough power to detect some small effect sizes.

MANUSCRIPT LOCATION: P19 L15

Reviewer #3: I had the opportunity to re-review the manuscript entitled " Latent transition analysis of cardiac arrest patients treated in the intensive care unit".

REVIEWER COMMENT: 1.Most of the revisions that have been suggested by me and the other reviewer's and editor has been incorporated in the manuscript.

RESPONSE: Thank you for the comments.

REVIEWER COMMENT: 2. The manuscript has been revised to not mirror it's previous study verbatim entitled ' Subphenotypes of Cardiac Arrest Patients Admitted to Intensive Care Unit: a latent profile analysis of a large critical care database'.

RESPONSE: Thank you for the comments.

REVIEWER COMMENT: 3. The sub-phenotypes of CA indicate a continuum of the disease process of CA and post cardiac arrest syndrome, the manuscript is not clearly demonstrating the clinical implications (not the further research implications) of the sub-phenotypes. The future research implication of the transition of the sub-phenotypes have been discussed in detail in this revision of the manuscript. 

RESPONSE: we add some lines to discuss the clinical implications of the sub-phenotypes.

RELATED REVISED MANUSCRIPT TEXT (or Table/Figure): 

The transition pattern of post-cardiac arrest syndrome is helpful for risk stratification, which is important for medical resource allocation and decision making. Furthermore, transition pattern might be indicative of medical interventions that can help to direct treatment, although current study failed to identify any difference in interventions across different transition patterns.

MANUSCRIPT LOCATION: P17 L20

REVIEWER COMMENT: Furthermore, There is a statement that is contradicting the conclusions and the clinical implications aspect of the discussion. The authors themselves state that ' Probably, the transition pattern is the intrinsic nature of the disease progression, there is no effective therapeutic intervention that could change the transition path' . but further go on to discuss the possible interventions that could be carried out at different sub-phenotype profile. This statement has to be modified or addressed in the discussion as it is contradicting the study aim.

RESPONSE: The statement has been modified in the revision.

RELATED REVISED MANUSCRIPT TEXT (or Table/Figure): 

Second, the study failed to find any interventions that were associated with transition patterns. Most probably, the fact that there was no difference in the medications between the patients that transitioned versus those that did not transition could reflect an intrinsic nature of the disease and lack of identifiable interventions that were associated with transitions. Other reasons are that the number of interventions being explored is limited and the study may not have enough power to detect some small effect sizes.

MANUSCRIPT LOCATION: P19 L15

---

## [Editor Report · Decision Letter 2]

14 May 2021

Latent transition analysis of cardiac arrest patients treated in the intensive care unit

PONE-D-20-40279R2

Dear Dr. Zhang,

We’re pleased to inform you that your manuscript has been judged scientifically suitable for publication and will be formally accepted for publication once it meets all outstanding technical requirements.

Kind regards,

Saraschandra Vallabhajosyula, MD MSc

Academic Editor

PLOS ONE
---

## [Editor Report · Acceptance letter]

18 May 2021

PONE-D-20-40279R2 

 Latent transition analysis of cardiac arrest patients treated in the intensive care unit 

Dear Dr. Zhang:

I'm pleased to inform you that your manuscript has been deemed suitable for publication in PLOS ONE. Congratulations! Your manuscript is now with our production department. 

Kind regards, 

on behalf of

Dr. Saraschandra Vallabhajosyula 

Academic Editor

PLOS ONE